# Disability Ghosting in the Double Lockdown Institution of COVID-19

**David Thomas Mitchell**

Department of English, Columbian College of Arts and Sciences, George Washington University, Washington, DC 20052, USA; dtmitchel@gwu.edu

**Abstract:** This paper surveys some of the voluminous journalistic coverage of the COVID-19 pandemic and the public health responses that ensued. While investigative reporting in newspapers and news programming played an important function, we expose the terms of the underreporting about the lockdowns in institutions for the disabled and elderly that ultimately changed little about public knowledge of the lives of disabled people who were always or already confined. Second, we detail the rapid unfolding of a critical journalism that revealed the mortality-dealing conditions of institutionalization beyond the acceleration of pandemic risk levels. Such governmental and for-profit run practices of letting individuals who were disabled or elderly die while in their care were enacted, of which residents could do nothing to protect themselves (in fact, risky exposure was a conscious practice of state governments during the unfolding viral epidemic). This essay argues, however, that a critical branch of COVID-19 journalism (largely based in the US) used investigative reporting to expose governmental miscounting, undercounting, and neglecting-to-count of disability deaths due to COVID-19 and/or to collect them under "other categories," such as the overall death rate of a population. Our key findings point out that despite the importance of this coverage, no one used this opportunity to talk with institutionalized disabled and/or elderly people—who were gravely at risk. Thus, we learned little about disabled peoples' lives as they were shipped back to congregate care settings and institutions from hospitals without treatment. An opportunity to explain disability institutionalization and its inherent dangers were lost despite the media saturation of coverage that rose in the wake of COVID-19 public health policies and practices.

**Keywords:** disability; journalistic representation; COVID-19; ghosting disability; viral journalism

## 1. Introduction: Reporting without an Other

Between March 2020 and late 2021, COVID-19 journalism exposed governmental and for-profit misreporting and neglect of disabled people warehoused in institutional settings (those which we refer to throughout as congregate settings of human warehousing). This vulnerable subpopulation was exposed and was excessively vulnerable to the ravages of the COVID-19 virus (and this was the major accomplishment of COVID-19 journalism in addition to its public health function as a disseminator of information). However, we arrived no closer to understanding how disabled people felt about their double lockdown situation (the first due to COVID-19, the second due to the fact that they were institutionalized and further isolated due to the risk of COVID-19 transmission, which happened anyway).

In general, these journalistic journeys out into investigative waters without cultivating information from disabled institutionalized residents themselves became increasingly evident as a feature of critiques of COVID-19 journalism. The neglect of institutionalized residents' perspectives will become "de Certeau-ian" through the ways we explain in the pages to come. Even the most critical journalism that exposed governmental neglect and bad practices that resulted in an escalation of deaths for institutionalized disabled people performed a parallel neglect of their own as they never managed to look behind the

windows of the institution to find out what was going on through disabled people's point of view.

As this survey and critique of COVID-19 coverage develops, it will become increasingly clear that the majority of coverage surveyed from the US (and predominantly from hub cities such as New York and Washington, DC) is where the coverage that exposed governmental neglect primarily occurred. In part, the US-centered nature of this report (although we also employ journalistic sources from the UK, Canada, and Mexico) is due to the fact that sequestration itself became increasingly part of the problem as a public health response; this was not in the usual terms it was reported, which were based on the stoppage of the economy and the ability to pursue profit as a right of the Western world, but rather in relation to the degree to which institutionalization became a petri dish that endangered disabled people forced into congregate care settings. It's important to acknowledge upfront that it proved exceedingly difficult for relatives to access reliable contact with institutionalized loved ones during the pandemic; however, since communication devices are so ubiquitous and include cell phones, institutional telephones, and the like, we press on the relative abandonment of media in attempting to speak to residents directly to assess their own baseline experiences.

## 2. Philosophy as Methodology

According to Michel de Certeau in his influential book, *Heterologies: Discourse of the Other*, the power of the text is parallel to the power of the social order in that they both compose and distribute places (labels and categories where one can expect to find recognizable categories of existence), thus all narrative and social organization involves a traversal and arrangement of space assigned and/or unassigned [1]. Within these composed and distributed spaces are objects-of-knowledge (institutionalized disabled people) positioned within those spaces in order to make them socially recognizable as unfortunate outcomes of suffering lives. This distribution of meanings to disability (those who are institutionalized due to round-the-clock care needs although also due to a lack of provision to live among others in neighborhoods of their choosing) fixes meanings of docility onto their unfolding dynamics and ever-transforming natures [2]. That which is bounded (i.e., assigned to its space) and devalued (i.e., frozen, at least for the time being, in meaning as an object) endures a kind of "social death"—a subjecthood who is "ineligible for personhood" [3]. Thus, for de Certeau, a proper "heterology of the Other" premises this fixing function of language as the primary violence of spatial/discursive partitioning between the observing subject (in our case the investigative COVID-19 journalists of disability and the pandemic) and the object of observation (patients/residents and body care workers in segregated congregate institutions ensconced within the parameters of the neoliberal for-profit nursing homes' walls).

Using our own neomaterialist "posthumanist disability studies" approach, we expose the terms of the underreporting regarding the foundations of the institutional lockdown that preceded COVID-19 and, unfortunately, changed almost nothing in the lives of those who were always or already confined [4]. Posthumanism has emphasized a return to the body orientation without the baggage of humanism, which has been compromised by taxonomies imposed on non-human flora and fauna. Such classification systems moved over to the human world and have also been imposed on racialized, disabled, and gendered subjects. Thus, posthumanists advocate for the junking of the "project of Western Humanism" altogether due to its biases of inferior people measured against a fully capacitated Euro-American whiteness, which proves exclusively detrimental to indigenous, enslaved, migrant, and indentured peoples. In turn, disability studies have argued against the false transhumanist position of overcoming the limits of humans due to the further tendency to exclude disabled body-minds altogether. Transhumanism argues that we must strive to make superlative corporealities and amendments by prostheticizing human forms into sci-fi-like beings (i.e., bionic eyes to replace 20/20 vision, butterfly wings on ambulatory bodies, etc.). Thus, our methodology that we refer to as posthumanist disability studies

promotes the sharing of knowledge as to what the navigation of non-disability accommodating environments can tell us about alternative ethical maps of interdependency that have to be pitched against the narrow norms of transhumanism. In effect, this is not what results from the insight of even the most radical COVID-19 journalism detailed below.

Both institutional residents and body care front line workers continued to interact with the latter staff population that went home to their families and communities of choice while disabled people remained under institutional lock –and key. The application of a neomaterialist posthumanist disability studies approach follows the material conditions that appear as given or natural, but prove, in fact, the failure of a constructed social world to account for a wider array of accommodations to allow more kinds of body-minds to participate in the fullest manner possible. We follow the mounting death toll and disposal of bodies to understand how a vulnerable population was impacted by sequestration within an already tightly defined confinement; however, our primary argument is that critical investigative journalism exposed the terms of state and for-profit discounting of disability death tolls, but failed to uncover anything substantial about why institutions are already the subject of a sustained disability critique.

Second, we detail COVID-19 journalism's exposé of the mortality-dealing conditions of institutionalization beyond the acceleration of pandemic risk levels against which residents could do nothing to protect themselves (in fact, risky exposure was a conscious practice of state governments during the unfolding viral epidemic). Finally, we examine something at the heart of COVID-19 journalism that revealed the predominance of pandemic deaths in nursing homes without surfacing with any direct account of the virally exposed victims. Thus, the overall effect of the journalism was to return with a discourse of the neglectful institutional and governmental refusals for accurate disclosures without any of the material disabled bodies in tow. Consequently, the journalistic methodology employed installed a "social and linguistic boundary" that was previously in place wherein disabled lives have something critical to offer non-institutionalized lives about the worlds in which they are radically confined and, ironically, reinstated the othering terms of institutional neglect and the pandemic that threatened to override its boundaries and give us a peek inside.

In a de Certeau-like manner of the exposé of the Other as a figment of the European travel writer's mirror projection onto the New World Other, a Covid pandemic journalism developed that ignored its own participation in the partitioning of the social and the textual. The COVID-19 journalism that emerged as a branch of mainstream journalism that covered the public health crisis in general—and the governmental recommendations for "keeping oneself safe"—is a discourse of the establishment. The institution could have been exposed as a fully malfunctioning and partitioned space; the terms of warehousing disabled humans in this manner further empties the affected subject of the right to its "own historical density" and the reader to the material conditions of such lives in which journalism presumably asks the social order to reinvest without cultivating any alternative specificity of what such revision might entail. Asma Abbas explains this approach as the way in which neoliberal orders pursue reparations claims as long as there is no necessity to hear the victims' story, to expose the social order to the particularity of suffering, and to channel all of its information about human experience into the quantifications of lost property [5].

## 3. Looking into the Mirror of COVID-19 Journalism

Perhaps the most consistent media image of the first half of the pandemic was the confined population of disabled people situated in their designated segregated spaces and confined from the liberties of movement, exchange, and intercourse [6]. This confinement is naturalized as a limitation within the bodies themselves rather than the product of controlled restrictions that are patrolled in a tightly controlled public space. While many of the particularities of the viral pandemic cannot be known—in fact, the pandemic is defined by its ability to elude predictable definition and place—as well as its timeline without apparent horizon and its variants already outpacing the efficacy of vaccines and boosters, the viral pandemic traverses spaces without any effective regulation and therefore qualifies

under de Certeau's definition of the *foreign*—that which escapes a place—in his influential chapter, "Montaigne's 'Of Cannibals': The Savage 'I'". De Certeau's analysis positions the Other as a mirror image of the fully capacitated, agential journalistic subject that is encountering the evacuated space of the Other (i.e., institutionalized disabled people held largely against their will in warehouses for humans). A world behind glass and/or locked entry doors and then mirrored by the locked doors of patient rooms that constitute the heavily portioned space of a majority of congregate settings. This is what we refer to as *the double lockdown of institutionalized disability*. Here, the media definition of disability defines the object of intervention through a lack of proximity that yields little information about the particularities of life experience but, ironically, fills that space with information that is abjected from the perceiver's point of view.

This process of projected abjection constitutes a space of the Other that appears to be rich, full-in-detail, and authorized through intimacies of interaction, but, in fact, is willed into being to the disadvantage of the object being abjected in space, time, and knowledge—a life caught up in the specularization of its death throes but disallowed from speaking its own truths. Its production as reiterative law or patterned repetition of degradation develops over time as naturalized and endemic to the object, thus it's attendant disavowal from subjecthood becomes synonymous with its story—empty vessels dying in confined waters further quarantined by media observers watching from the shorelines of history (i.e., outside on the sidewalks of the institution or over the phone with institutional administrators and government officials). This form of distanced coverage occurs without any actual interaction that might accumulate in a situation of intimacy of reportage. Thus, journalists tended to bring their assumptions about disability to the glass windows, beyond which they could not penetrate. What was left was a degraded, vulnerable, and unfortunate subject without a subjectivity to probe.

We want to argue that Michel de Certeau's analysis of sixteenth century travel narratives regarding first contact stories with racialized native peoples applies to disability as an overdetermined object of inevitable mortality by COVID-19 in the time of a global pandemic (late 2020–late 2021). The pandemic travel narrative constructed around escalating COVID-19 deaths of individuals in nursing homes and other double lockdown structures of congregate sequestration ironically erupts out of a similar gap created through the appearance of journalistic immersion in the site of the institutional sequestration. This study is by no means exhaustive, nor does it claim to cover the large global expanse of COVID-19 journalism with regards to the pandemic in the US (as well as some of the coverage in the UK, Canada, Mexico, etc.). Instead, we attempt to hone our sights on spaces of non-interaction (an absence that is the content of disability in the nursing home outbreaks of the COVID-19 virus) with a disability Other that was (and continues to be) reified by the journalistic coverage as dying, dead, or already absented. We confine ourselves mainly to newspaper and online newspaper reporting and primarily analyze mainstream coverage that appeared in the UK and North American publications, such as *The Guardian, BBC News, El Universal, The New York Times*, and *The Washington Post*, as these have been the predominant sites of encounter for our analysis of what we refer to as COVID-19 journalism. Finally, we limited our search to the primary COVID-19 pandemic period of publications that stretch from March 2020 to the end of October 2021 (the key period of transmission, multiple outbreaks, and mortality due to lack of vaccination, the accelerated circulation of variants, and the movement of unvaccinated body care workers into and out of the space of nursing sequestration).

According to a *New York Times* article, nearly one-third of all coronavirus-related deaths in this period are linked to nursing homes [7]; whereas, disabled residents have comprised an enormous percentage of deaths in the overall death rate of the virus, body care workers—primarily low wage people of color who function as low remunerated caregivers in neoliberal Western economies—also comprise a key aspect of the institutional death rate. As Leah Lakshmi Piepzna-Samarasinha puts it in *Care Work: Dreaming Disability Justice:* "This is for everyone Black and brown who freeze, who feel we could never, ever

think about asking someone to do our dishes or clean our toilet or help us dress, because that is the work we or our families have done for little or no money during enslavement, colonial invasion, immigration, and racist poverty" [8]. As part of Piepzna-Samarasinha's radical reformulation of the intersection of disability and racialized body care workers is a third, overlapping factor added to the mortality toll outlined here, which includes those with disabilities who are also people of color.

## 4. Ghosted Lives

Yet, what of the object that is partitioned into a space that consists for the media as little beyond the fact of its sequestration. By approaching disability in this manner, journalism prevents its audiences from knowing about the nature of the experience of that isolation and its attendant limitations of movement, liberty, choice, exposure, risk, etc. In these cases, we come upon a "ghosted entity" that is already prepared for death in its role as "bare life" or sacrificial scapegoat. "Bare life," in Agamben's terms, forces a question, "[w]e must instead ask why Western politics first constitutes itself through an exclusion (which is simultaneously an inclusion) of bare life. What is the relation between politics and life, if life presents itself as what is included by means of an exclusion?" [9]. The ghosted are those who occupy this "cultural location of disability" as this in-between space of those who are included by way of their very exclusion [10]. This inside-out designation proves necessary to assure overseers—the experiencers/recorders/administrators of the pandemic—that valuelessness is primarily what is being invested in the object so that the lives of the valued can continue relatively unfettered with the least amount of disruption. This approach affords the ghosted residents of institutional life risk aplenty while an ultimate kind of exclusion of disability persists at its foundation. This is how we pursue the readings to come in relation to disability as the object of COVID-19 investigative reportage's object *par excellence*, yet without any essence of engagement of the lives inside the institutional-lockdown-ironically-turned-pandemic-quarantine-space.

Our argument in this essay is that the nursing-home-interred were one of the most catastrophically impacted populations of the ongoing COVID-19 pandemic and that the representative function during this imbalance of death was actively observed and reported on by mainstream and "other stream" journalism, which was intended as furtherance to the machinations of the neoliberal state and for-profit institutional practices. Thus, importantly, the investigative reportage politicized the space of the nursing home as a death trap of pandemic transmission. Yet, in this investigatory mode that endeavored to expose institutions, nursing homes, group homes, and other lockdown facilities populated by elderly, imprisoned, and/or disabled people alike, journalists focused on the institution as one of the primary, disregarded vectors of viral contagion transmission without accessing the stories and lives of the disabled people experiencing its death-dealing impact. Journalists talked with neighboring residents; health care workers; police, fire, and emergency technicians; first responders; institutional administrators; and the private, for-profit investors who own these facilities, yet left what we refer to as a ghosted population of disabled, dead, dying, and at-risk subpopulations (the primary affected object of institutional epidemics) completely absented from the conversation.

The logic of not having access to those confined within the nursing home industry and its variants played out as common sense by journalists who could only see their reflection of disability as long-term, undesirable suffering. One can imagine the COVID-19 journalist narrating the terms of this lack of direct interaction in some version of the following argument "since they are under double lockdown quarantine, we (the journalistic corps) cannot gain access to them so we're getting as close as we can through stories of the professional classes set out to oversee and administer their lives." Neoliberalism bars the direct interaction with its confined populations while claiming to offer them up for reportage-by-proxy. The affected proved unavailable within the recesses of the institution in which their social death preceded their actual deaths.

Within this return to what Foucault in *The History of Madness* refers to as "a dialogue without [a patient].," disability becomes a distance forged but not crossed [11]. Forged in the sense of the fact that *journalists dipped their toes into the water of the institution, waded into the death-making viral transmission factory of the neoliberal, for-profit, private nursing home, and stepped back out of the stream with an incredibly important story of disability as a sacrificed, scapegoated subpopulation whose mounting death toll was ghosted from view*. Yet, in that effort to forge the distance that the modern day nursing home represents between readers and the disabled population confined therein, no interaction with a disabled person emerged; no portrait of lives radically separated from others; no exposé of the institution's new moon of contagion despite never having been a safe harbor prior to that moment in the "long durée" of institutionalization from the Middle Ages forward [12]; and no dialogue with the object of what Polish author of psychiatric killings in Poland during World War II, Stanislaw Lem, referred to as *Szpital Przemienienia/The Contagious Hospital* (1982) [13].

There were photos and videos galore of institutional residents behind glass and plexiglass but almost no example of an interview, phone conversation, non-auxiliary discussion with anyone other than *representatives* for those existing in the now viral death mills of post-industrial institutional wards (Figure 1). *As in de Certeau's critique of the European travel narrative, disability was relegated to a non-space of interaction while appearing as the savage object with whom everyone was conversing.* The dead and dying were already gone, so-to-speak, as their status among the disappeared of the COVID-19 pandemic had already been accomplished by something akin to Cachia's formulation of a "social death" controlled through their defining "ineligibility of personhood" that was constituted in their pre-pandemic lives [14].

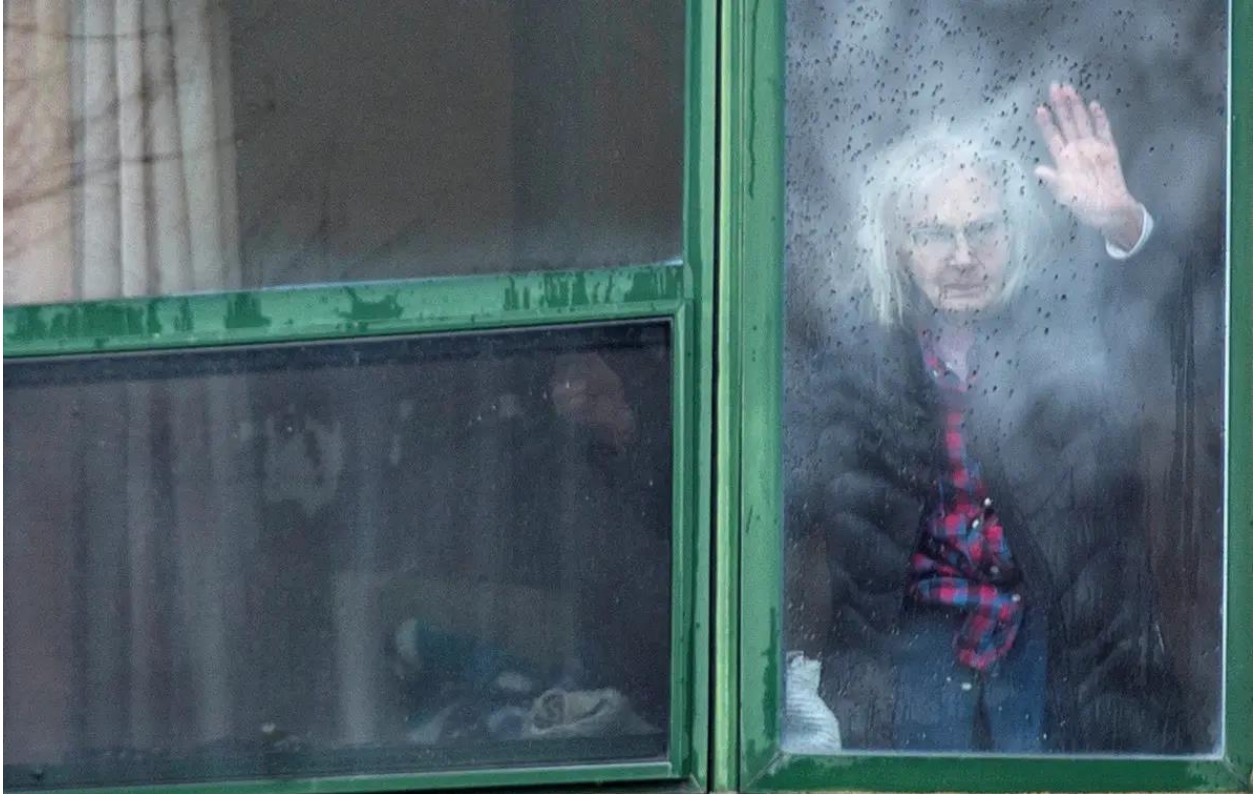

**Figure 1.** Photo credit: Christinne Muschi [15].

## 5. The Euphemistic Object of Ideology

Disability proved a difficult term to define precisely as it was consistently buried along with other bodies in a vague and unspecified manner in mainstream journalism. Within this obfuscatory logic, disability was sometimes referred to more euphemistically under

the umbrella category, "the elderly," who comprise a majority population in one space that is marked by both its institutional boundedness and its narrative space of reportage, wherein the subject that has surfaced during the COVID-19 epidemic is absent. "The elderly" functions as a euphemistic descriptor in that it calls up incapacities that have been naturalized across an illusory normative lifecycle rather than those institutionalized based on qualifications of "burden of care." While we do not intend to argue that "the elderly" was not a space designated and occupied within the journalistic coverage of the pandemic, the category was comprised of forms of "non-normative being" that were far more fraught than appeared on the surface—which is based solely on age [16].

For example, Fordham University bioethicist, Charles C. Carmosy, referred to nursing homes as part of American consumerist "throwaway culture" [17]. Most appropriately, Carmosy identified the hodge-podge of ghosted institutionalized residents as comprising "the reality of cognitive impairment, aging, and death" that contemporary US culture cannot embrace "forthrightly with the moral and social equality of every human being." This interweaving of disability, age, and the dying that Foucault referred to as the "strangely mixed and confused" subjects of the eighteenth-century asylum in "The Great Confinement" continues in the incongruous practice of lumping residents in contemporary nursing homes comprised of those who share status as unproductive or insubstantially productive [18]. Thus, "nursing home", like "asylum" before it, has always engaged acts of rhetorical triage that cover over the illogical range of variation it contains and the nature of those who are affected by neglect, abuse, violence, and social and familial abandonment. However, perhaps, even more accurately, the strange brew of residents exposes the illogic of imprisonment overall and its lack of definitional logic cannot be found because it does not exist outside of those diagnosed and sentenced as non-normative others that are institutionally disallowed from sharing public space. The institution is an after-effect of what Judith Butler and Athena Athanasiou theorize as the formative nature of "dispossession", which has led to a radically devalued subpopulation's consignment: "subjects deprived of the ability to have control over their life, but they are also denied consciousness of their subjugation as they are interpolated as subjects of inalienable freedom." Disabled nursing home residents are, within this formulation and to riff off of Slavoj Žižek's most infamous work, the euphemistically sublime objects of ideology [19].

This dual denial of control over life and denial of consciousness of subjugation come together in the mutual constitution of nursing home subjectivities as a key site of pandemic sequestration. The intersection of these denials form the reflection that is the basis of the "de Certeau-ian" mirror of COVID-19 journalism. The convergence of pandemic breakout and journalistic reportage began in the late fall of 2019 and continues into the current moment of fall 2021 (the time of the writing of this analysis). In our survey of journalistic coverage since that time there are at least three major shifts to note in the reportage: the first involved the tracking of the outbreak within nursing homes and mounting death tolls serving as the proverbial canary-in-the-coal-mine of unchecked viral transmission in the lockdown wards; in other words, COVID-19 journalistic coverage of the public health threat amounted to a warning about how the virus was on its way out of the labyrinth of congregate settings to the "real world" [20]. The second was a spate of reporting on insufficient or non-existent PPE protocols practiced by staff in nursing homes since the outbreak of the pandemic and indicative of a lack of medical training by "body workers", who are predominantly low wage working people of color [21]. The third shift involved pushing COVID-positive patients back into nursing home populations from hospitals or those who were never evacuated to hospitals for appropriate medical care in the first place [22]; furthermore, a closely linked series of branch reporting within the third shift involved a recounting of the retrospective death tolls [23,24], the meting out of lawsuits [25], and the toppling of policies from the loosening of the anarchic-era-of-nationalist deregulation and civil liability practiced among the for-profit, private nursing home capitalists and administrations [26].

Confined within what proved most striking at first glance from our initial listing of this four-stage series of transitions in reporting is that the mass media privilege stories

centered on gentrification, staff insufficiencies, stealthy outbreak quantifications, and the disruptions of the lives of for-profit, nursing homeowner-operators. The third wave of coverage foregrounds the pushing back of nursing home residents to institutions and might suffice as a site where reportage, pandemic, and those unduly subjected to the pandemic might be expected to ultimately meet. Yet, even in stage three, this meeting rarely happened, and the nursing home pandemic populations were journalistically narrated as a reflection from the outside looking in. Such developments are framed by Asma Abbas' argument regarding the "usual weapons" that liberalism wields in hiding away the subjects of suffering: "the separation between the public and the private and other comparable dichotomies, the centrality of the agent, and the neat economics of emotions and sense experience" all merge to move any tangible knowledge of suffering based on experience off the stage of reparations capitalism. Liberal humanism is only fitted to the restitution of property to the neglect of the value of lives, or even the value of knowledge that might be gleaned from an actual exchange as to the particularities of life that went on in COVID-19 lockdown institutions.

### 6. First Shift: Suffering without Sufferers

First-wave mainstream COVID-19 coverage was rife with these tactics and ways to cover suffering without an encounter with the ontological nature of the suffering and/or the sufferers themselves. On 7 March 2020, the euphemistically named "Life Center of Kirkland" in Washington state became a key "hot spot" for the disease as reported by *Washington Post* reporters, Sacchetti and Greene. The article detailed how the "coronavirus quietly spread" and that the mounting illness and death toll impacted people over age 60, "particularly those with lung or heart disease or some other conditions." Thus, the article made surface the Foucauldian "strange mixture" of nursing homes cited above as elderly people qualified by disabilities and difference, which was contained by a normalized integration of disability into the human life course. The report then went on to detail the myriad disruptions that the virus meant for the staff, administrators, and even neighbors, quickly leaving behind any residue of the elderly disabled people who were initially identified as the primary at-risk population in the pandemic. Sacchetti and Greene, two of the earliest journalists to begin covering COVID-19 fallouts in institutions, detailed a bevy of colliding coordinates caught up in the pandemic viral transmission dragnet, including rising real estate prices accompanied by "fast-paced demographic changes" that made long-time residents wonder if gentrification was going to make trailers on front lawns inappropriate, skiing trips of top executives end abruptly, or the local fire chief and his family cancel their planned Alaskan cruise. The most telling detail, perhaps, was that advertisements had gone out in local media about the center's new hiring push for nursing staff at $17.50/hour.

In other words, the drama of the "Life Center of Kirkland" nursing home (along with its intact neoliberal luxury living language of middle class decadence) as the ground zero of the pandemic in the US was narrated through a sharp separation between public and private, the centrality of the agents whose work and leisure lives were upended by the outbreak, and the neat economics of emotion that kept the reader and family members of those dying inside while talking to them on the phone and peering through a glass window at their suddenly more-than-acceptable-at-risk relatives on display for families to see even in the aftermath of their decision to sequester them in the institution. In other words, the story was told as one of unexpected risk for contracting COVID-19 while leaving behind the narrative of how the patients came to be there in the first place. Thus, journalistic coverage acted from the start in a fully "de Certeau-ian" manner by journeying to the land of the strange and exotic Other without understanding the culture of the institution in which they found their lives bound. In fact, the strange irony existed where the institution was already a confined and segregated space even before COVID-19 arrived with its mandatory or highly recommended lockdown protocols.

This partitioning of nursing home residents (positive or negative) would come to dominate the ghosting imagery of the pandemic for the foreseeable future. This exclusion without details of the exclusionary institution in which the viral outbreak first surfaced would dominate and stand in for—even literally reflect—de Certeau's key formulation of the gap that exists between the active observing self and the projected passive space of the Other. Accompanying headlines in this early representational period crept up to the glass wall in equally stealthy ways as a discontinuous pathway to those infected or potentially infected with the virus. BBC.com reported that residents at the Somerset Commissioning Group in the UK told residents they suddenly all needed to file "DNR [Do Not Resuscitate] orders" [27]. *The Guardian* reported on 27 March 2021 that the outbreak was exerting an "invisible death toll" on nursing home residents [28], and the *Washington Post* detailed how thousands of nursing home residents were being killed by COVID-19 due to not being moved to hospitals for care [29]. Overall, the opening period of reportage led to a conclusion similar to that announced by the mainstream Mexican newspaper, *El Universidad*, on 7 June 2020 that deinstitutionalization advocates had been arguing for centuries: "Nursing homes are a ticking time bomb" [30].

Yet, while de Certeau's argument posited a relatively simple dichotomy between observing subject and mirroring object, COVID-19 coverage set up a slightly more obscure pathway to reportage. As discussed above, reporters plumbed administrators, staff, first responders, and others of the middle and upper managerial classes for news on the outbreak and its increasing spread. Thus, reporters themselves were at least one more step removed from the "de Certeau-ian" formula of the Other in that they neither talked with nursing home residents who were rapidly becoming the most visible at-risk community for the mounting death toll that now began to stand-in for all citizens (particularly those in North America). The nursing home residential Other was not a mirror-image of the observer's abjected desires, but rather such desires were fed through what Asma Abbas calls the "media sensorium" by comments from family members and, most often, the middle-class professionals given over to the management of their disabled subjects in the eugenics era of the late nineteenth and early twentieth centuries [31]. As the age of normalization took shape and disabled people were increasingly assigned a team of overseers to implement rehabilitation schemes that would ultimately promise to erase their differences and return them to the norm from which they had fallen (the unrequited promise of nearly all sequestering institutions), their images would be increasingly refracted through the prism of those who were not synonymous with subjectivities forged in the crucible of direct experience. So went the representation of disability in hearsay accounts that filled up the COVID-19 reportage of the pandemic.

One could argue that this prism of reflected surfaces leading to the tangential indirect encounter with the nursing home's disabled Others could merely be credited to the highly transmissible nature of the disease. Even reporters have to be safe and shield themselves from risks of transmission. Those who reported on the "unfolding mystery" surrounding the rapid spread of the disease through nursing home populations to the general population outside the walls seemed capable of garnering reports of communication that circulated widely and with apparent relative ease. Families received daily updates from loved ones ensconced within the nursing home facility by telephone on the other side of their glass partitions, and nurses reached out with reports of good news then almost immediately followed up with concerns about the sudden onset of symptoms, such as respiratory difficulties arriving out of nowhere and sudden deaths to those who appeared fine earlier in the day. Whereas the "de Certeau-ian" direct witness ventured out into the space of the Other and returned with little more than his (sic) own fears and aspirations projected into the gap of the space of the Other, the COVID-19 pandemic reporter refused to chase down a more direct line of ways to know the situation of those inside the walls. As a primary result of this circuitous lineage of representation, the nursing home Other remained her/himself a mystery, shielded behind walls, separated from the public, and suffering within an interior space that could not be breached by any other; those who have the power to represent

without surrendering their socially assigned roles as gatekeepers to the subjective lives of the interred.

## 7. Second Shift: The Other behind Glass

This development of a mystery pursued during the transition to the second shift of COVID-19 reporting morphed into a hall of mirrors. An image of lives filtered through intimacies that reflected, deflected, and partitioned communicative lines so that consumers of media about nursing home impacts would keep alive institutionally dependent economies that were threatened while shielding audiences from opportunities for more direct interactions. As Abbas again phrases the problem: "These puzzles (the nature of our knowledge of suffering that are rarely known through the suffering agent) revolve around, ironically, the agents qua respondents to someone's suffering—the saviors, liberators, lip-readers, empowerers—whose regard for others is fed by a fervor steeped in the unacknowledged privilege of framing these conversations and puzzles and is subsequently quite taxing to those who suffer." Steeped in this theory one can readily interpret medical advertising trends of our own historical moment as that which directs viewers to physicians, nurses, first responders, psychologists, and even lengthy catalogues of potential mortality-dealing side-effects without ever holding a direct exchange with the perspective of patients on behalf of whom such advertising is presumably directed. Ghosting disability, it turns out, is a product of middle-class professional obscurantism as the logic of the value of the congregate disability institution coming home to roost.

What came to dominate the narrative of COVID-19 in the nursing home during the second shift in reportage was an even more dramatic transformation to a story told predominantly through stories of health care worker "ineptitude;" there were chronic nursing home problems, such as understaffing, inadequate access to PPE, poor medical training, employment of an underclass that would work for near minimum wage, and the body care workers own mounting death tolls [32]. Late May 2020 signaled this new tactic in representations of COVID-19 when *The Washington Post* exposed that "major nursing home chains violated federal standards" of infection control and prevention of transmission by nursing home staff and reported on fines for nursing homes that were mounting due to "coronavirus lapses" [33]. The fact that a majority of body care workers and front-line responders to public health crises are people of color and that their welfare along with the welfare of the nursing home resident (who was also often a person of color) went unrecognized for so long points to journalistic exposés of pandemic spread punctuated by a relative lack of investment in the nature of the mortality-recipients' experience of risk. This state of affairs increasingly turned the pandemic into an affair of what post-colony theorizer, Achilles Mbembe, refers to as necropolitics. Necropolitics is a sequestration of devalued subjects in a space that directly reflects their social invalidity. The mortality-dealing conditions of such a space are well recognized as "secondary outcomes" of death, as those ensconced are already socially dead [34].

This spate of reports about nursing home worker "lapses" in protection became the epicenter of reporting in the second stage as inadequate PPE, lack of oversight, and infections from the outside world brought into nursing homes fueled a second and third wave of outbreaks. The journalistic target of such recurring outbreaks held out body care workers as a new vector in the COVID-19 and delta-D variant transmissions. A *New York Times* story released on 9 May 2020 pointed to the fact that after nearly one year into the pandemic spread, one-third of all COVID-19 deaths were among residents and institutional health care workers [35]. Almost eleven months previously *The Washington Post* recorded the 10,000th death in the area nursing homes of DC, Maryland, and Virginia; three months later a similar report identified that "nursing homes fell short on preventing infections" [36]. In other words, the nursing home industrial complex of the US had gone from a ravaged site of transmission to what Henry Giroux refers to as neoliberalism's defining feature: a scapegoating that "destroys all vestiges of the social contract, and increasingly views 'unproductive' sectors—most often those marginalized by race, class,

disability, resident status, and age—as suspicious, potentially criminal, and ultimately disposable" [37]. Nursing homes as the source of these various lapses were deflected onto their human caregivers and away from the owners of for-profit institutions, which became increasingly shielded during the second stage of COVID-19 reporting. Not only had the Trump administration destroyed key regulations that made nursing homes more culpable for outcomes, but the nursing home industry itself increasingly took refuge behind immunity laws [35]. What came to the fore of COVID-19 nursing home reportage was the degree to which the liberal exchange of caring for the least valued citizens (this included racialized nursing home staff, first responders, and front-line workers interacting with the public) came with the caveat of protection from liability. While nursing home residents and workers interacted behind glass windows and concrete walls, the nursing home industry became increasingly *immune* to shouldering responsibility for its managed populations [38].

## 8. Third Shift: The Institutionalization Industry Moves Culpability to the State

Around the same time that the nursing home industry moved into its immunity from responsibility for the death of residents in its care, COVID-19 reportage also began its transition to a third stage of coverage regarding the return of nursing home residents from hospitals despite continuing infection. This third shift centered on the Cuomo administration's secreting of true death tolls and its strategic dumping of nursing home patients from hospitals back into the sites of their initial transmission contact point. Luis Ferré-Sadurni opened queries into coronavirus deaths by asking "How Accurate Is New York's COVID Death Toll?". The article detailed how New York and other large states such as California and Texas were allowing a lower death toll to be recorded based only on tests performed by a coronavirus test lab. This methodology deviated from more reliable counts reported by the National Center for Health Statistics, which used death certificates submitted to state health departments. By undertaking this contrast Ferré-Sadurni found that the official New York death toll of 43,000 fell significantly lower than the 54,000 deaths compiled by the Center for Disease Control. Governor Andrew Cuomo defended the practice with a peculiarly circular logic: "We have always reported lab tested COVID results. That's what our reporting has always been. He added that the CDC asks for additional information 'which we report to them, and they report" [39].

The circularity of the argument is important as it returns us to the story of the return in de Certeau's analysis of the production of the space of the Other. After "encountering" the Other on its own turf, so to speak, the travel writer completes a circle that returns with the knowledge gained by the interaction to the homeland from which s/he departed. Along the way, the travel narrative digests the information to traverse the vast space of the return, thus emphasizing the literality of the distance that marks the journey. This process of the journey out, witnessing of the barbaric Other, and the return home occurs as a witness who reports on what he has observed directly and thus carries the reliability of first-hand knowledge. Thus, the space of the Other is reported as if the process of coming to know the Other through direct interaction is indicative of the effort and spatial traversal of the venture, which turns out to be little more than a projection of fears, desires, and the otherness of the narrators encountering their "abjected not-I" [40]. The text, as de Certeau phrases it, creates a space of traversal that never leaves home while seeming to engage in a substantive understanding of difference: "But the written discourse of the Other is not, cannot be, the discourse of the other.". Thus, the travel writer enters into a productive void that enacts the exchange yielded by a non-existent intimacy—the pursuit of the Other "induces" the writing, yet functions as an absent center that cannot be filled in because familiarity with the Other is what the story lacks. The story of the Other appears as a gift to eager audiences that have never left their own shores. This receipt of the tale of the Other appeases a desire to "cannibalize" the Other along with the narrator's own ingestion of the story beyond the walls of the social order he has vacated and now returns to deliver as the fruits of his labor.

However, New York Governor Andrew Cuomo's administration would accept going down in the blaze of accusations by female employees that they experienced sexual harassment on the job in the governor's mansion and downtown offices. In a truly Baudrillardian situation, the scandal that undoes his leadership is sacrificed to cover over the true scandal of COVID-19 mass deaths [41]. That Cuomo had cultivated a sexist culture and used his power to manipulate female employees resulted in a legal finding of a misdemeanor of "aggressive contact." But, in truth, the reality of the Cuomo administration's fall can be found elsewhere in the concealment of and consignment of institutionalized disabled people to deaths from the COVID-19 pandemic. His leadership's trafficking of nursing home patients back to the institution before any successful treatment endangered everyone in the nursing home and the disabled COVID-19 patients themselves. Exposures in the double lockdown of COVID-19 institutions resulted in mass deaths and the imbalance of mortality experienced by residents, particularly prior to the availability of a vaccine regimen. The Cuomo administration was fully aware of this mortality-dealing practice and the governor's resignation occurred, one might easily speculate, in order to avoid charges of nursing home exterminations.

The third shift coverage further branched off from the return of COVID-positive patients through a retrospective re-counting of death tolls, reports on nursing home staff's reluctance to get vaccinated, the meting out of lawsuits, and the toppling of policies from the loosening of the anarchic-era-of-nationalist deregulation and civil liability by for-profit, private nursing home capitalists and administrations. COVID-19 journalism began with articles about nursing home staff being the least likely to be vaccinated, as reported by Reed Abelson in *The New York Times* on 16 September 2021, and then was dominated by reporting about scrambling mandates for health care workers in medical and care-taking industries to get vaccinated [42]. Otterman and Goldstein reported that staff in New York hospitals were to be released from their jobs as they "spurn[ed] vaccines." Expectations were that workers would quit in droves, mass firings were on the horizon [43], and a shortage of home health care workers might result from punitive measures taken within the institutions of care and medicine. In other words, the risk of COVID-19 institutionalized residents was left behind in order to report on vaccine requirements that would result in the abandonment of nursing home patients.

Further reports revealed administrative considerations to replace unvaccinated health care workers with the National Guard, workers spurning vaccines and therefore endangering hospitals [44], decisions regarding mandating staff vaccines or not getting paid for institutional care [45], and new COVID-19 outbreaks amidst calls for staff to be vaccinated. As the COVID-19 nursing home scandal deepened, reports became increasingly more defensive of the nursing home industry by alarming readers of the vaccines coming mutiny by health care professionals. The foregrounding of workers and institutional decision-making demonstrate the degree to which the lack of reconciliation with nursing home residents' perspectives continues to be withheld from the "official" pandemic narrative as told by the US media.

## 9. Revelations from the Space of the Foreign

As in de Certeau's analysis of Montaigne's essay "Of Cannibals", the representatives of the Other reveal a host of ironic observations about the operations of national institutions that are regarded as superior to the cultures within which they are in charge. The revelations from the space of the foreign prove to entirely upset that which had become naturalized inequality within the post-colonial narrative of life as we know it. In this manner, perhaps, de Certeau offers a way to account for the end run around nursing home residents' perspectives that characterize the investigative reporting of the COVID-19 pandemic. As "The Savage 'I'" explains, "For these socio- or ethno-cultural boundaries to be changed, reinforced, or disrupted, a space of interplay is needed, one that establishes the text's difference, makes possible its operations and gives it 'credibility' in the eyes of its reader, by distinguishing it both from the conditions within which it arose (the context)

from its object (the content)." Thus, COVID-19 journalism that exposed nursing homes as viral death traps posited a "context" (for-profit institutions as transmission zones) and differentiated the spaces of exposé as distinct from "its object" (any intimacy with those disabled and/or elderly patients dying from their location in the double lockdown institution). The reporting pursued a social critique of institutions as endangering their residents, yet set tight parameters on the location, time, and duration of the object's endangerment. Interviews with the Other never surfaced, therefore allowing COVID-19 journalism its political edge with little evidence of direct knowledge about the suffering object presumably under examination.

One article published in the *New York Times* by Danny Hakim held out some promise for this longed-for perspective from disabled residents impacted by COVID-19. The headline read, "'It's Hit Our Front Door': Homes for the Disabled See a Surge in COVID-19" [46]. The article drops down through a series of perspectives awaiting the arrival at the subject referenced in the "our" of the title: first, a comment by the Executive Director of the non-profit CP Nassau network of care facilities; two parents of developmentally disabled people in Brooklyn were next cited; then, three state employees who are direct care aides were quoted; following that, the building manager of the Lexington Park Condominium (which turns out to be a residence for nursing home patients) comments on the lack of information about COVID-19 transmissibility from state agencies; Marco Damiani, chief executive of one of New York's private service providers, comments on the nature of institutions in his network that are hardest hit; and, finally, Jennifer O'Sullivan, spokeswoman for the state agency overseeing the residences, comments on the state of emergency all institutions in her oversight are currently in. The "our" of the title is difficult to find but actually arrives in a quote from the manager of residences at Northwell who has issued orders for nurses to go in and test the residents: "It was real. Oh my God, this is real. It's hit our front door." Disability in the COVID-19-confined space of the Other turns into phantasms of others' comments about the ravages experienced by the nursing home industry itself as opposed to those within its care. No perspective of the residents ever surfaces and thus the "our" becomes a form of obliteration of the ostensible subject.

## 10. Conclusions: Subject to Return

Once the travel narrative returns home with the observing self's reflections on its encounters with the Other, the Other is already inexorably lost. The distance travelled is great, the obstacles to the journey out and those that hinder a safe return are numerous, and the whole passage ventured appears to retain the hallmarks of actual interactions with the exotic Other that is its reported object of recovery. The Other is a number, a loved one, a patient, or an unfortunate casualty of a nursing home resident or staff worker who happened to be in the wrong place at the wrong time. The entire oeuvre of COVID-19 journalism of the nursing home in the pandemic functions akin to de Certeau's exposé of Montaigne's parody of the sixteenth century travel narrative. In Michel de Montaigne's essay, "Of Cannibals", which set the "de Certeau-ian" theory of the Other into motion, Montaigne exposes the lack of facticity of the direct witness' report with a visit by three visitors from the New World brought back on the ship from New France (i.e., in this case sixteenth-century indigenous people of Brazil) [47]. The New World visitors of Montaigne's essay become the first direct encounter with the fictionality of the Other by the reader. The "savages" from the space of the Other remark on three key observations that they discuss in an interview with the narrator [presumably Montaigne's narrator-manqué] about what they think of the luxuries of Paris as opposed to the barbaric nature of their lives back in South America: (1) they are surprised that the gap between rich and poor is so great and that the more numerous poor do not rise up and take the horded wealth of the rich who manufactured the inequitable social order that has led to their economic dispossession; (2) they are equally surprised that adults take orders from a child-king (Charles VIII who ascended the throne at the age of 13 after his father's death) who ruled France in the late sixteenth century; and the third thing they identify is forgotten by the narrator.

There are many candidates for what this missing third critique might prove to be as Montaigne never fills in the gap; most likely the empty lost commentary is the essay's pretense at filling in the gap that exists between the teller and his lack of intimacy with the object of the tale. The French authorities expect expressions of awe and marvel from these New World visitors [the Others} and get social critique instead. Montaigne's exposé shows that the gap between self and Other is an abyss of miscomprehension without literally asking those who are being represented. This is how we analyze COVID-19 pandemic reportage as a journalistic return that appeared on all surfaces as an exposé, a helping narrative that opened-up the closed system of the nursing home industry to the invasiveness and hard questions of investigative reporting. Yet, one remains struck by how little this roughly two-year stint of reporting managed to fill in the gap that separates readers from the confined disabled Other. The nursing home as a contemporary version of Lem's "contagious hospital" and lockdown institution with its coterie of administrative professionals emerged as the content of the pandemic's subject and the subject of the pandemic's scourge. The Other takes the shape of a body bag filled but zippered closed for viewing.

Unlike the pandemic and to a much greater degree than the confined non-nursing home population, the institution of the nursing home and other congregate settings feature a double lockdown. First, the twist of the lock on the doorknob in nursing home rooms where sequestration is not a feature of the pandemic as it is for others, but rather already a bounded space distributed for disabled people who require assistance and who presumably cannot be cared for in their own homes. The second double bolt on the entry door of the nursing home resident involves the lockdown of exposure, of lacking say or command of one's space and an inability to set a schedule of when individuals may enter, leave, perform rounds, deliver, and dispense medications, etc. Each uncontrolled entry brings in the world and potentially subjects the nursing home internee (for what else can we call them in this space of the double lockdown?) to exposures without liberties to risk. Thus, the sequestration is of an institutional nature with employees, staff, and visitors with varying levels of mobility to retreat to their own confined home spaces during the time of the pandemic while disabled people remain confined in a space originally designated for them as a bounded, finite space with highly restricted movement and a hospital-like loss of privacy—a space in which the confined are locked down, pre-determined, fixed in their location, collectively grouped and differentiated only to the extent of a bag of billiard balls [48]; their valuelessness is on double lockdown and disability is the *foreign* that cannot escape its place.

How do we understand the reasons behind the fact that even in the underreporting of disability deaths such mortalities amounted to nearly 40% of the accountable total? What does it tell us about the nature of these institutionalized ghosted lives? For example, in New York where it turns out the total is likely greater than 50% and that the infections were largely the result of nursing home employees provisioned with inadequate PPE who moved in and out of the institution to their places of chosen sequestration (i.e., apartments and homes) and yet went largely untested for COVID-19 by their employers. How do we deal with the fact that states forced nursing homes to re-admit patients diagnosed with COVID-19 from hospitals, failed to isolate them from other institutionalized residents, and stopped testing all together in order to avoid delivering accurate figures because they would have revealed the state was "selecting" nursing home patients in what Foucault in *Discipline & Punish* theorizes as the biopolitics of those allowed "to take life or let live?" What does the revelation that for-profit nursing homes were re-admitting COVID-19-positive residents despite knowing they lacked staffing levels, appropriate supplies of PPE, and the expertise to adequately care for them mean? What does it mean that residents were returned to a key vector of viral transmission while being "relocated" from hospitals? Here "relocated" took on a meaning akin to Jewish "deportation" in World War II as having one's citizenship stripped away followed by a train car ride across a border that effectively rendered one "stateless" and, therefore, body-minds and rights became inaccessible at the hands of a

carceral state and nursing home industry that maneuvered to make such dispossessions more material. Thus, as Judith Butler and Athena Athanasiou provocatively ask, "how has the human been formed and maintained on the condition of a set of dispossessions?"

Yet, there is also a discursive determinant to these journalistic reports that sought to expose underreporting of deaths in nursing homes as a revelation of systemic neglect. Not only were deaths officially underreported to deflect attention away from the open graveyard that nursing homes have become (and, likely, always have been), but the reporting itself performs its own ghosting function. Marx premised that the further one went into the abstraction of value, the further one moved away from the material conditions of production [49]. While reporting on nursing home cases of COVID-19 related deaths, the journalism that sought to expose establishment neglect (even in some of the most progressive states, such as New York, California, etc.), failed to return in their reports with any details of the lives of the victims. This practice continued even as it became a staple of electronic news media to provide examples of lives lost to COVID-19.

The institution borrows techniques from the prison and the psych ward borrows its isolation and incarcerating practices from the prison and policing systems intended to limit the participation of non-normative bodies. The institutions of the police state are all working together importing and exporting techniques to increase power over body-minds deemed unacceptably deviant. In the case of COVID-19, we find institutional systems that argue for their practices as benign in that caretaking is their specialty and, in neoliberal speak, the product they deliver to disposable people. Yet, masking carceral networks as caretaking institutions keeps disabled and elderly people who need care as those who must exchange their liberty to receive supports. What this three-tiered evolution of COVID-19 journalism failed to make clear is how institutionalization itself exposes vulnerable populations to risks that are endemic to every lockdown setting. Thus, the over-representative death toll of institutionally sequestered subjects surfaced as a key exposé of COVID-19 journalism.

However, the problem this essay tries to demonstrate is that the difficulty of getting to institutionalized residents during COVID-19 placed their experiences of carceral space overall as tangential to the reporting. Thus, readers could infer the risk of transmissible mortality in a viral epidemic as a general property of institutionalization itself, but the reporting unintentionally kept the problem as exclusive to this moment of viral exceptionalism—a situation of a highly transmissible disease ravaging carceral spaces. COVID-19 accelerated a symptom of institutionalization, and journalism never grappled with disability perspectives in order to discover that transmissibility is a recognized side effect of carceral existence itself. Thus, in our minds, COVID-19 mortality was one more nail in the coffin of the inadvisability of congregate institutionalization itself.

While one cannot interview the dead, COVID-19 journalism abandoned the perspectives of victims, their families, or those currently in nursing homes awaiting the next outbreak while pursuing a spectacular governmental exposé. The journalistic text reports the figures, and this act anticipates the arrival of a proximity to the lives of nursing home residents, yet the figure that is the institutional ghost never returns with the journalistic narrative of devalued material bodies and their conditions of existence. This absence of the material voices of those affected by nursing home exposures, in other words, performs a second-level discursive ghosting of the victims and future victims and their intimates by showing that the writing about the deaths did not include those who were marked as its victims and thus COVID-19 journalism cannot be *their* discourse.

**Funding:** This research received no external funding.

**Institutional Review Board Statement:** Not applicable.

**Informed Consent Statement:** Not applicable.

**Data Availability Statement:** Not applicable.

**Conflicts of Interest:** The author declares no conflict of interest.

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
