# Peer review of "Disability Ghosting in the Double Lockdown Institution of COVID-19"

_societies, doi:10.3390/soc13070170_

Round 1

Reviewer 1 Report

I struggled with this paper. I found it difficult to read and tried to imagine one of the people the authors thought should have been heard during the pandemic reading it. They would have given up in despair. The authorial thesis is that reporting of COVID in nursing homes failed to foreground the voices of residents. This does not surprise me. I knew people in such places and accessing them, even though they were friends, was incredibly difficult.  This is acknowledged, but inadequately. I also note that although you purport to cover journalism in several countries the overwhelming coverage is of the US, even New York. fair enough except you claimed to do more.

My recommendation is that you rewrite this in plain English, review your coverage of the different countries’ press that you claim to represent, and try again.

I apologise for late review. 

Author Response

I'm sorry that this reviewer felt the findings were disappointing to disabled people who died of covid in congregate settings. I found them sad as well. However, this is not the thesis of the paper, but rather that a critical journalism of investigative reporting used their coverage to expose governmental miscounting, undercounting, and simply ignoring the counting of disability deaths due to covid. In order to clarify that this journalistic branching occured in the coverage I will simply refer to it as "covid journalism" with a discussion of permutations in coverage and the branching of the field during the two primary years of the epidemic: i.e. when it happened and why it happened in the way that it did while leaving behind institutional residents' points of view.  Our key point of discovery is that despite the importance of this coverage no one used the opportunity to talk with institutionalized people who were gravely at-risk. As the reviewer expresses, "this is not surprising," however, this is not the principle argument. Rather the problem occurs in the ghosting of disabled lives that represent merely a reflection of journalistic imaginings and thus left a relatively empty shell as its ostensible objects of rescue. Nor did we learn about disabled peoples' lives as they were shipped back to congregate care settings and institutions from hospitals without treatment.  Thus, government misreporting and neglect of this vulnerable subpopulation was exposed (and this was the major accomplishment of covid journalism in addition to its public health function as a disseminator of information), but we arrived no closer to understanding how disabled people felt about their double lock-down situation (the first due to covid, the second due to the fact that they were institutionalized and further isolated due to covid transmissions which happened anyway). This journey out into investigative waters without getting information from disabled institutionalized residents themselves becomes DeCerteau-ian in the ways we detail as even the most critical journalism that exposed governmental neglect and bad practices that resulted in an escalation of deaths for institutionalized disabled people, performed a parallel neglect of their own as they never manage to look behind the windows of the institution to find out what was going on through disabled people's point of view. I will make it more clear that the majority of coverage surveyed is in the US and predominantly from the NYT as this is where the coverage that exposed governmental neglect occurred. I will openly recognize the difficulty of reaching disabled institutional residents during this time as the reader points out. 

Reviewer 2 Report

Dear Authors, your manuscript is original, but I find the methodological gaps in approaching the concept insurmountable. I do not find the introduction of “disability” appropriate, and describing " the glut of territory seized by viral journalism " is anachronistic

Other concerns

The authors are missing

7 But what do you mean by counter covid-journalist? journalists to counter disinformation? It is not clear.

15 I don't understand, the concept of text power coupled with a model of displacement.

22 I suggest being less clear-cut in the connection between social disposition and the collapse of subjectivity

52 “Excusatio non petita, accusatio manifesta “

instead of sequestered population, I would suggest confined. (throughout the manuscript)

59 double “controlled”

68 the double-lock down of institutionalized disability… I don't understand how you generated a definition of this magnitude, without even the semantic description of disability

72 reference missing

73-75 no reference.. it seems to me that the custodial nature of a public health measure has been emphasized. Underestimating the salvific urgency of confinement.

82 why?

87 to cover the glut of territory seized by viral journalism ???

Author Response

The definition of disability is wide ranging and not limited to any specific catalog of diagnoses (usually institutionalized disabled people have multiple impairments); however, the essay makes the definition of disability more dependent on the fact of institutionalization for those in need a chronic care. This definition which blurs elderly and disability populations assumes centralized, round-the-clock medical care is needed. Our primary argument is that disability results in forms of non-normativity that require human warehousing as a normative response to withdrawal from participation in social networks of life that are necessary to human well being. This second point of the article argues that institutionalization is a petri dish with vulnerable people made even more vulnerable. However, whatever one thinks of institutionalization even the most critical journalism that exposed governmental and institutional misreporting, undercounting, and neglect of counting deaths, all failed to find out from institutionalized residents what they were experiencing. This is not simply because they were hard to reach, but rather, as we argue, covid journalism (that which undertook a radical critique of a massively over-represented vulnerable subpopulation) never looked behind the glass to find out what disabled people were experiencing. This was what we theorize as particularly deCerteau-ian about the covid period in that even the most governmentally destabilizing journalistic coverage never assessed from institutionalized residents perspectives. Thus, this population was ghosted despite the massive level of coverage (they were in the mix, counted among the dead with various obfuscations in the general lumping practices of deaths in the larger population, but not consulted as to their experiences during this public health emergency). We will clarify the what we mean by counter-covid journalism by referring to it as a branch of the prolific nature of coverage overall. The misunderstanding of the power of the text to displace is the basis of de Certeau's formula of heterologies which we discuss for 3 pages in our explication of his critique of travel writing that we make parallel to counter-covid journalism. We will change "sequestered" to "confined" as suggested. We will be less clear-cut about the relationship of social disposition to the collapse of subjectivity. Point 72: disability is usually used to mean mass while impairment is aligned with specific kinds of non-normativity. We explain how we'll clarify this decision as outlined earlier in this response. We will fill in the two missing references identified in 72, 73-75. The critique of "underestimating the "salvific urgency of confinement" is, respectfully, not the point. Everyone except frontline workers were confined, but even the most critical wing of covid journalism did not ask disabled people about whether they found "salvific urgency" as a justification for being locked down in the institution. This is not a critique of covid-based sequestration as a governmental policy of safeguarding public health; it is a critique of a critical aspect of the coverage developed by covid journalists. The essay will clarify how these developments effectively furthered an epistemology of pwds' exclusion as an indication that an ‘able-bodied’ person’s self-signification of not being ‘other’—the disabled-- resulted alongside of the exposé of governmental intentional undercounting. The wall between disability and able-bodiedness on the journalistic end was never effectively breached. I will add this explanation into the paper so any potential misreading can be managed better. Viral journalism is journalism that overtakes the space of newspaper and social media and therefore appears to be covering every nook and cranny of the impacted social world; territory is seized when disabled persons' perspectives are shuttled aside. I will explain this matter by inserting the sentence -- or something similar -- above into the paper.

As to the line-by-line edits I have repaired as identified below:

Dear Authors, your manuscript is original, but I find the methodological gaps in approaching the concept insurmountable. I do not find the introduction of “disability” appropriate, and describing " the glut of territory seized by viral journalism " is anachronistic 

*Fixed: defined disability in institutions more explicitly and removed "glut of territory" phrasing

Other concerns

The authors are missing

7 But what do you mean by counter covid-journalist? journalists to counter disinformation? It is not clear.

*fixed: changed to covid journalism with various permutations defined as first, second, and third stage coverage tactics

15 I don't understand, the concept of text power coupled with a model of displacement.

22 I suggest being less clear-cut in the connection between social disposition and the collapse of subjectivity

*Fixed: removed phrasing altogether

52 “Excusatio non petita, accusatio manifesta “

*This is an analysis of the coverage not a manifesto although I am not adverse to manifestos when the squandering of peoples' lives are at stake

instead of sequestered population, I would suggest confined. (throughout the manuscript)

*Fixed: changed all instances of sequestration to confined

59 double “controlled”

*Fixed: removed

68 the double-lock down of institutionalized disability… I don't understand how you generated a definition of this magnitude, without even the semantic description of disability

*Fixed: This is the point of theory in cultural studies to take a case and make an argument of magnitude. You can feel dissatisfied if you want, but the Humanities does not depend upon quantitative surveys of magnitude to make larger claims. This is it's freedom to not be penned in by sociological restrictions and scientific expectations of "proof."

72 reference missing

*Fixed: there was no reference required as they are words that summarize journalistic coverage of covid in general

73-75 no reference.. it seems to me that the custodial nature of a public health measure has been emphasized. Underestimating the salvific urgency of confinement.

82 why?

*I don't know what this "why" refers to, but give me some context and I will do my best to respond

87 to cover the glut of territory seized by viral journalism ???

*Fixed: re-wrote the entire sentence to sound less colonialist in its maneuverings

Reviewer 3 Report

Dear editor and Author, I received this manuscript to review but it is far from my knowledge.

I consider the topic very interesting, I am sorry, but I am not able to give you a proper review because is far from my background

best regards

sara santilli

Author Response

At least this reviewer who recused themself said they found the topic very interesting.

Reviewer 4 Report

The paper presents a study on the the terms of the underreporting on the foundations of the institutional lock-down that changed little  about public knowledge of the lives of disabled people who were always-already sequestered. The topic of the paper is interesting and of potential impact on society development. However, the paper requires major changes: the literature review section needs a more robust and coherent theoretical framework about posthumanist disability studies. The methodological section lacks in clarity. 

Author Response

I will try to clarify the interpretive framework using DeCerteau's methodology and insights in her influential book, "Heterologies: Discourse of the Other". I will either further define neomaterialism as an aspect of posthumanism or cut it out altogether. 

*Fixed: I have rewritten the opening in clearer language and explained neomaterialist posthumanism more thoroughly in the methodology section. The literature review section (not to mention the entire paper) is quite substantive as it moves in-and-out of more than 70+ newspaper and media articles about the topic covered. Not to mention it uses a substantive theoretical grounding in Foucault, Mbembe, De Certeau, Montaigne, Abbas, Mitchell and Snyder, Giroux, Zizek, Marx, Butler, Anthanasiou, Braudel, Agamben, Piepzna-Samarsinha, Cacho, Antebi, etc.

Reviewer 5 Report

I have just a general comment on the overall paper that is relevant to this special issue on media and social representation. This a very good and interesting work base on the the De certeau framework  and his heterotical methodology of the Other in order to provided a different  perspective on covid 19 pandemic treatment, and, in particular disability described as the Other by reporting journalisms.  The use of the cripepistemological methodoly  in posthumanist  disability studies perspective contribute to reveal of discursive process of the social construction of the ederly and the disabled as the Other.

Author Response

This reviewer has actually read some of the key materials discussed in this paper. Unlike others who have not been adequately exposed to this area of investigation via deCerteau and why it is fully appropriate to the critique of covid journalistic coverage of the over representation of institutionalized residents in the mounting death toll that governments and institutions intentionally tried to hide. Andrew Cuomo effectively resigned from his office as NY governor on a sexual harassment charge when the more serious issue was institutionalized residents escalating death counts as he forced them to return to the double lock-down of institutionalization. This was akin to hiding a mass murder by taking the lesser charge of sexual harassment as his indecency. Thanks to this  reader for understanding the nature of the philosophical line of argumentation applied to the critique of social and news media coverage that never breached the wall to find out what disabled people knew about the world of the institution and its death-dealing effects on those who found themselves confined (twice).

Round 2

Reviewer 1 Report

Thank you for the revised paper. You responded energetically to my comments. It is easier to read now, and the US bias is acknowledged. And yet, the argument feels tangential. If your purpose is to reveal the way institutional residents are 'ghosted', you do a reasonable job, but I would expect some reference to longer term issues. It has, as you acknowledge, long been the case that people in institutions are invisible, out of sight out of mind, and 'othered'. For instance Goffman's ideas about institutional care are extremely relevant. And I return to the argument that accessing nursing home residents during the pandemic was incredibly difficult, reliant on over-worked staff who themselves may not have been proficient in use of video conferencing etc. I tried to run a history project with people who live in residential care during the pandemic. So often, staff left people struggling with the phone or ipad, or just decided that some other activity was more important, and we could not communicate with them.  So hard to blame journalists. I'd like to see the long term nature of this invisibility emphasised more in your paper. It is publishable now, but could have been more consistently argued.

Author Response

In regard to longer term issues of institutionalization and its petri-dish like conditions for vulnerable populations we have added this paragraph: 

The institution borrows techniques from the prison and the psych ward borrows its isolation and incarcerating practices from the prison and policing systems intended to limit the participation of non-normative bodies. The institutions of the police state are all working together importing and exporting techniques to increase power over bodyminds deemed unacceptably deviant. In the case of covid we find institutional systems that argue for their practices as benign in that caretaking is their specialty and, in neoliberal speak, the product they deliver to disposable people. Yet, masking carceral networks as caretaking institutions keeps disabled and elderly people who need care as those who must exchange their liberty to receive supports.  What this three-tiered evolution of covid journalism failed to make clear is how institutionalization itself exposes vulnerable populations to risks that are endemic to every lock-down setting.  Thus, the over representative death toll of institutional sequestered subjects surfaced as a key exposé of covid journalism.

However, the problem this essay tries to demonstrate is that the difficulty of getting to institutionalized residents during covid placed their experiences of carceral space overall as tangential to the reporting. Thus, readers could infer the risk of transmissable mortality in a viral epidemic as a general property of institutionalization itself, but the reporting unintentionally kept the problem as exclusive to this moment of viral exceptionalism — a situation of a highly transmissible disease ravaging carceral spaces. Covid accelerated a symptom of institutionalization and journalism never grappled with disability perspectives in order to discover that transmissibility is a recognized side effect of carceral existence itself. Thus, to our minds, covid mortality was one more nail in the coffin of the inadvisability of congregate institutionalization itself.

Reviewer 2 Report

D

Author Response

We have clarified the research questions, made the argument more succinct while also offering a concluding two paragraphs that explains what these findings demonstrate with respect to the future of institutionalization. There are 71 citations to back up the findings and argument of this essay and they all pertain explicitly to relevant to the journalistic coverage we are analyzing. 

Reviewer 4 Report

With the revisions made, the manuscript is clearer and meets the criteria for the publication in a scientific journal. The argumentations are well substantiated and the research methodology is grounded. 

Author Response

Thanks for your earlier suggestions and your support on this final revision of the manuscript.